# Reducing workplace burnout: the relative benefits of cardiovascular and resistance exercise

Rachel Judith Bretland and Einar Baldvin Thorsteinsson

Department of Psychology, School of Behavioural, Cognitive and Social Sciences, University of New England, Australia

## ABSTRACT

**Objectives.** The global burden of burnout cost is in excess of $300 billion annually. Locally, just under half of working Australians experience high levels of occupational burnout. Consequently, burnout interventions are paramount to organisational productivity. Exercise has the potential to provide a multilevel and cost effective burnout intervention. The current study aims to extend the literature by comparing cardiovascular with resistance exercise to assess their relative effectiveness against well-being, perceived stress, and burnout.

**Design.** Participants were 49 (36 females and 13 males) previously inactive volunteers ranging in age from 19 to 68 that completed a four week exercise program of either cardiovascular, resistance, or no exercise (control). Randomised control trial design was employed.

**Method.** Participants were measured against the Subjective Exercise Experience Scale, the Perceived Stress Scale, and the Maslach Burnout Inventory.

**Results.** After four weeks of exercise participants had greater positive well-being and personal accomplishment, and concomitantly less psychological distress, perceived stress, and emotional exhaustion. Cardiovascular exercise was found to increase well-being and decrease psychological distress, perceived stress, and emotional exhaustion. Resistance training was noticeably effective in increasing well-being and personal accomplishment and to reduce perceived stress. The present findings revealed large effect sizes suggesting that exercise may be an effective treatment for burnout. However, given a small sample size further research needs to be conducted.

**Conclusion.** Exercise has potential to be an effective burnout intervention. Different types of exercise may assist employees in different ways. Organisations wishing to proactively reduce burnout can do so by encouraging their employees to access regular exercise programs.

Corresponding author
Einar Baldvin Thorsteinsson,
ethorste@une.edu.au

## BACKGROUND: BURNOUT, PSYCHOLOGICAL STRESS, AND MULTILEVEL INTERVENTIONS

In a study covering America, Asia and the Middle East, *Golembiewski et al. (1998)* found that 60% of public sector and 40% of private sector employees reported high levels of burnout. Burnout is understood to be a chronic state of job stress, where employer expectations and employee workload exceed the individual's perceived psychological

capacity and ability to cope with the work demands expected of them (*Maslach, Schaufeli & Leiter, 2001*; *Mutkins, Brown & Thorsteinsson, 2011*). The global burden of burnout through decreased productivity, retention, absenteeism, and compensation costs in excess of $300 billion annually (*Rowe, 2012*), as such the World Health Organisation is predicted to report burnout as a global pandemic within the next decade (*Nash, 2013*).

Psychological stress describes an interaction between an individual, their work environment, and other external influences (*Malone et al., 1997*). Psychological stress is understood to occur when there is a substantial *perceived* imbalance between demands placed on the individual, and perceived response capability, and is aggravated in situations where failure to meet the demand has perceived adverse consequences (*Malone et al., 1997*; *Thorsteinsson, Brown & Richards, 2014*). Effective coping is then measured based on emotional reactions and cognitive functioning in response to the stressor, in a fluid process dependent on evolving work demands and a fluctuating personal outlook (*Lazarus, 2000*; *Malone et al., 1997*). *Lazarus (2000)* articulates that this is a cognitive-motivational-relational theoretical concept of *appraisal* based on individual differences (both intra- and inter-individual), whereby an individual constructs relational meaning from the person-environment relationship, social and physical influences, personal goals, self-belief, and available resources and subsequently makes an assessment of ability to cope based on his or her perceptions of demand compared to capability.

Job-related psychological stress, was first termed *burnout* in 1975 (*Maslach & Jackson, 1981*). Burnout can be described as prolonged exposure to occupational pressure including emotional and interpersonal stressors (*Maslach, 2003*). Burnout comprises three central components: emotional exhaustion, depersonalization, and (lack of) personal accomplishment (*Maslach & Jackson, 1981*; *Maslach, Schaufeli & Leiter, 2001*). Of the three, emotional exhaustion is the most noticeable and often the primary symptom (*Golembiewski et al., 1998*; *Meesters & Waslander, 2010*). Emotional exhaustion can also be viewed as depletion of an individual's emotional resources, typically characterised by statements such as "I feel emotionally drained from my work, and used up at the end of the workday" (*Childs & Stoeber, 2012*; *Maslach & Jackson, 1981*). It is proposed that emotional exhaustion underpins burnout and through the coping mechanisms employed by the emotionally exhausted individual, the other burnout elements depersonalization and reduced personal accomplishment arise (*Harwood et al., 2010*; *Maslach, Schaufeli & Leiter, 2001*). Depersonalization can be seen as a detachment barrier, where the emotionally exhausted employee introduces emotional and cognitive distance between them and their situation in an attempt to cope with their workload (*Cordes & Dougherty, 1993*). This shields the individual from emotional strain but results in a dehumanized perception of others. This is often accompanied by cold disregard for the needs of clients and callous indifference towards clients' feelings (*Maslach, 1986*, p. 4). Continuation in the depersonalized state leads to a callous and cynical outlook and generalised indifference to the organisation (*Maslach, Schaufeli & Leiter, 2001*).

As emotional exhaustion and depersonalization progress, the burnt out individual feels guilt and inadequacy about their emotional limitations which leads to a reduced sense of

personal accomplishment (*Maslach, 1986*). An individual's perception of their ability to excel and perform worthwhile tasks is diminished and they no longer feel of value to the organisation (*Golembiewski et al., 1998*; *Maslach, Schaufeli & Leiter, 2001*).

As *Maslach (2011)* highlights, there has been a high degree of interest in interventions targeting burnout, and these may occur either on an individual or organisational level, sometimes referred to as *micro* and *macro* approaches, respectively. Individual interventions often involve removal of the burnt out individual from the workplace and can be effective in reducing emotional exhaustion, but tend to leave the interpersonal and occupational elements of depersonalization and personal accomplishment unaltered (*Maslach, 2003*). As noted by *Halbesleben, Osburn & Mumford (2006)* individual programs do little to change the workplace environmental stressors, therefore not addressing the underlying cause of burnout, subsequently proving to be ineffective overall. By contrast, departmental occupational based interventions can be applied through managers to departments or workplaces as a group (*Maslach, 2003*).

Consistent with the realisation that individuals are nested in organisations and are intertwined with social interactions, research in burnout interventions has evolved to consider multi-level interventions that focus on improving management practices at the team level as well as workplace health practices on an individual basis (*Maslach, Leiter & Jackson, 2012*).

It is in this domain that exercise interventions are an appealing proposal. Whilst commitment to an exercise program occurs at an individual level and provides individual results, departmental initiation and encouragement of team members is likely to enhance the effect of the intervention and promotes workplace involvement, motivation and provides social opportunities for employees to interact. In addition, corporate exercise programs are cost efficient, saving organisations an average of $4 in healthcare costs for every $1 invested (*Shusterman, 2010*).

## LITERATURE REVIEW: THE EFFECTS OF EXERCISE ON PSYCHOLOGICAL STRESS AND BURNOUT

The concept of exercise and its positive impact on mental health is not a new phenomenon. Dating back to the 1970s research has reported that exercise results in increased mood, self-concept, and work performance including greater productivity and reduced absenteeism (*Seraganian, 1993*). Even though there are clear health benefits of exercise, there appears to be a void in the research connecting it with burnout, with relatively few reported studies of which most have been cross sectional correlational analysis that assess the relationship between "exercise," "physical activity" or "leisure activity" and "stress" (*Gerber & Puhse, 2009*). A shortfall of the literature, and astutely highlighted by *Gerber & Puhse (2009)* in their analytic review, is that detail regarding the type, intensity, and duration of exercise involved in the studies is limited and often oversimplified.

A review of the effect of exercise on positive mood states was conducted by *Berger & Motl (2000)* who found that high intensity exercise was optimal for cardiorespiratory and metabolism benefits, but had little impact on desirable changes in mood or positive

well-being. In contrast, moderate intensity exercise was shown to have sub-optimal fitness benefits but was consistently associated with positive well-being and mood benefits (*Berger & Motl, 2000*). More recent research by *Cox et al. (2004)* has challenged these findings and found the two exercise intensities comparable. In subsequent research, *Cox et al. (2006)* utilized the Subjective Exercise Experience Scale (SEES), which measures positive well-being, psychological distress, and fatigue. They found that vigorous exercise had a greater effect on positive well-being than moderate intensity exercise, whilst concomitantly finding an overall reduction in psychological distress in comparison to a control condition, but with comparable efficacy between the two treatment conditions (*Cox et al., 2006*). Whilst the aforementioned literature compared moderate and vigorous cardiovascular exercise, thereby preventing direct comparison, the current cardiovascular condition would be considered a vigorous training program, whereas the resistance training program is moderate intensity. Also, *Cox et al. (2006)* found that whilst participants reported increased fatigue during the exercise, 30 minutes post-exercise fatigue had returned to baseline measures. As highlighted by *Hecimovich, Peiffer & Harbough (2014)*, fatigue and exhaustion are often poorly distinguished. However, consistent with their distinctions, the current study views fatigue as an acute physical condition. In contrast, this study views exhaustion as chronic, psychological, emotional exhaustion as defined in the burnout literature (*Maslach, 1986*).

One recent pilot study ($n = 12$) has assessed the impact of cardiovascular exercise on burnout and perceived stress, finding a significant result with large effect sizes against emotional exhaustion ($d = 1.84$) and depersonalization ($d = 1.35$) but not personal accomplishment ($d = 0.31$) (*Gerber et al., 2013*). *Gerber et al. (2013)* also found a significant effect of cardiovascular exercise in reducing perceived stress using the Perceived Stress Scale (PSS). Both *Gerber et al. (2010)* and *Ben-Ari (2000)* report that the stress buffering effects of exercise increase with increased frequency of participation in exercise, suggesting a relationship exists between frequency of exercise and perceived stress, well-being, and burnout gains. *Thorsteinsson, Brown & Richards (2014)* suggest that high work-stress may increase anxiety, depression, fatigue, and organisation staff turnover; and reduce organizational commitment and job satisfaction; leading to adverse work outcomes for organisations and employees.

## PURPOSE OF THE CURRENT STUDY

Introducing exercise to a sedentary population and measuring the effect over time (four weeks) of different exercise types (cardiovascular and resistance) against multiple measures of stress including well-being, perceived stress, and burnout has not been attempted. Research in this area will deliver valuable insight into understanding how exercise impacts perceived stress and burnout. Organisational burnout is the result of a unique combination of stressors specific to each workplace, therefore a single burnout intervention is unlikely to be universally effective. This research aims to provide insight into which aspects of exercise are effective in reducing the different aspects of psychological stress and burnout to allow exercise professionals, individual employees, and managers to tailor employee exercise

programs to the exercise type, frequency and duration most applicable to the burnout as experienced by that particular employee.

It was hypothesised that (a) the exercise conditions (resistance and cardiovascular together) will show increased positive well-being, decreased perceived stress, and decreased burnout health state after four weeks of exercise; (b) cardiovascular and resistance exercise will affect positive health state and perceived stress and burnout differently; and (c) exercise frequency and duration will be positively correlated with positive health state and negatively correlated with perceived stress and burnout health state.

## METHOD

### Participants

Ethics approval was obtained from the University of New England Human Research Ethics Committee (approval number HE13-051). Participants were 58 volunteers from the Tamworth and Armidale locality (Males $n = 15$, Females $n = 43$) ranging in age from 19 to 68 ($M = 36.79$, $SD = 13.51$). Nine participants were excluded from the study due to: underlying medical conditions (unable to gain medical clearance to exercise) ($n = 5$), failure to complete the program ($n = 3$), and for already conducting significant exercise (3 or more hours per week) ($n = 1$). Participants were randomly allocated to three conditions: control ($n = 20$), cardiovascular exercise ($n = 20$), and resistance exercise ($n = 9$). This left the study with 13 males and 36 females (total $n = 49$) ranging in age from 20 to 68 ($M = 36.63$, $SD = 13.46$). Figure 1 represents diagrammatically the flow of participants through the study.

Participants' job status was similar, with all but three participants being employed or studying for a minimum of 20 hours per week. Those participants not employed for a minimum of 20 hours per week, were excluded from comparison on the Maslach Burnout Inventory (MBI).

### Measures

Participants completed three self-report measures of stress and well-being, demographic questions, a health screen based on SportUNE pre-exercise medical standards, and an exercise diary as detailed below.

**The Subjective Exercise Experience Scale** (SEES; *McAuley & Courneya, 1994*) measures the positive and negative poles of psychological health, as well as subjective indicators of fatigue in response to exercise. The SEES is a 12 item, one-word measure of emotion on a 7-point Likert scale: 1 (*Not at all*), 4 (*Moderately so*) and 7 (*Very much so*). The SEES has no global measure but comprises of three subscales: Positive Well-being (e.g., "terrific"), Psychological Distress (e.g., "awful"), and Fatigue (e.g., "tired").

The SEES has been assessed for convergent and discriminant validity against the Positive and Negative Affect Schedule (PANAS) with positive well-being correlating with the positive affect scale ($r = .69$, $p < .01$), and with psychological distress correlating with negative affect ($r = .41$, $p < .01$) (*McAuley & Courneya, 1994*). The fatigue subscale was unrelated to positive or negative affect (*McAuley & Courneya, 1994*). The SEES has strong internal

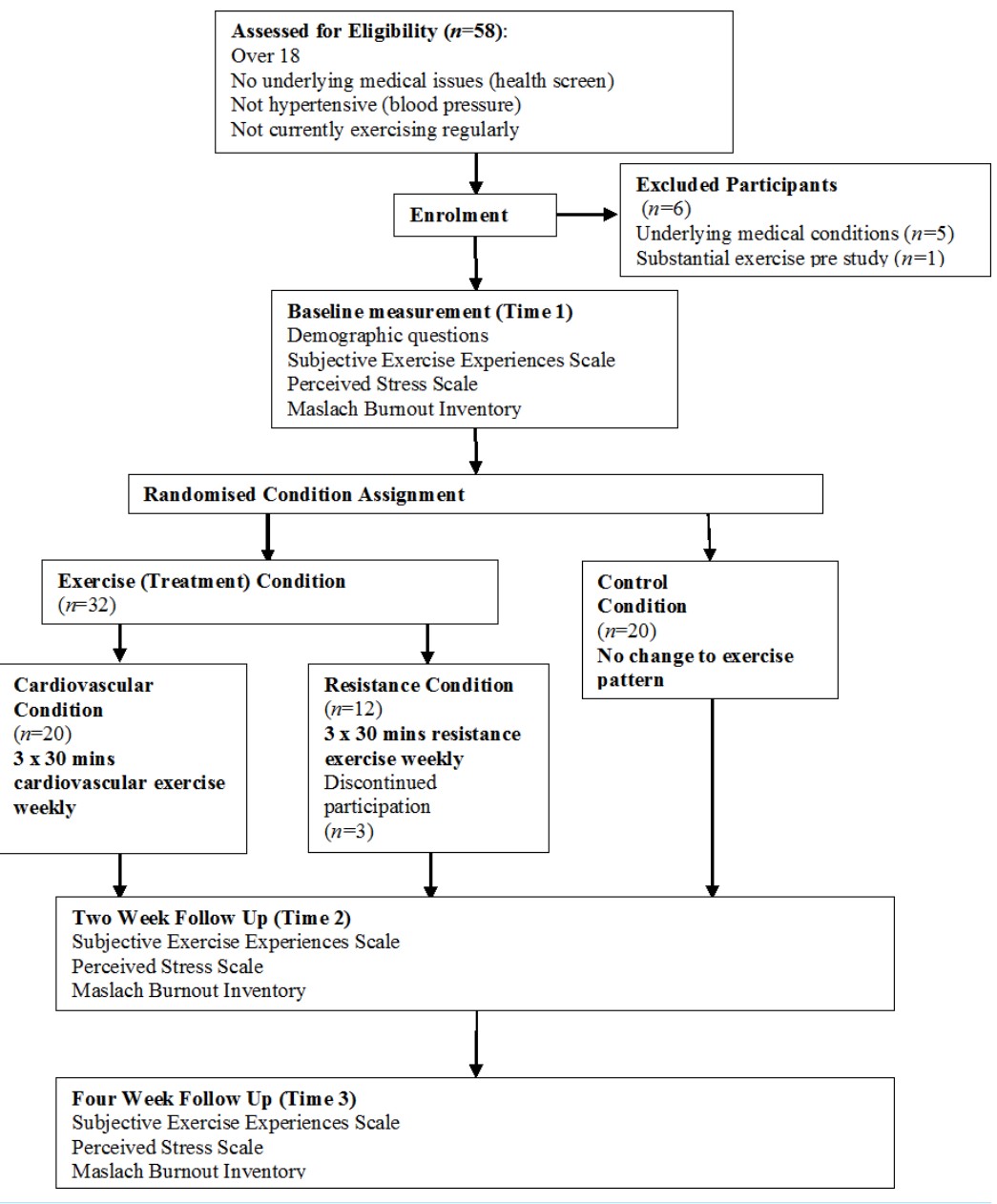

**Figure 1 Schematic overview of study design and measurement times.**

consistency positive well-being .86, psychological distress .85, and fatigue .88 (*McAuley & Courneya, 1994*). Cronbach's alpha in the present study were strong: positive well-being .89, psychological distress .74, and fatigue .88. The SEES was chosen for this study given its design to allow comparative analysis before and after exercise measuring both the increase of positive affect and the reduction or absence of negative symptomatology (i.e., anxiety, depression) (*McAuley & Courneya, 1994*). Additionally, this scale provides a subjective measure of physical fatigue, which for the present study provides a useful comparison to the measure of emotional fatigue (i.e., the emotional exhaustion subscale of the MBI).

**The Perceived Stress Scale** (PSS; *Cohen, 1994*) is the most widely used instrument for measuring the degree to which situations are perceived as stressful to the individual over the last month. It is a 10 item instrument with each item rated on a 5-point Likert scale from 0 (*Never*) to 4 (*Very Often*) (*Cohen, Kamarck & Mermelstein, 1983*). Items 4, 5, 7, and 8 are positively stated (e.g., "In the last month, how often have you felt that things were going your way?") thus reverse coded for analysis. The PSS has strong internal consistency with a Cronbach's alpha of .85, and test-retest reliability of .85 (*Cohen, Kamarck & Mermelstein, 1983*). The PSS has been tested for convergent and divergent validity against the State-Trait Anxiety Inventory-Trait Version ($r = .73$, $p < .001$), and the Multidimensional Health Locus of control ($r = .20$, $p < .001$) (*Roberti, Harrington & Storch, 2006*). In the present study, Cronbach's alpha was excellent at .89. Whilst other self-report measures such as the Social Readjustment Rating Scale (*Holmes & Rahe, 1967*) are available to measure a cumulative stress score based on a list of stressful life events, these scales assume that the measured events are consistent across individuals and are the precipitating cause of pathology and illness (*Cohen, Kamarck & Mermelstein, 1983*). For the current study we do not assume that response to stress is consistent, as we know that response to workplace stress varies with levels of burnout (*Maslach, 2001*). Therefore the PSS is a more suitable measure of perceived psychological stress as it does not assume consistency of individual response to stress triggers, and it measures stress without reference to specific life events, making it an appropriate choice given the current burnout focus (*Roberti, Harrington & Storch, 2006*). As previously introduced, situations where perceived demands are in excesses of physical or personal resources are commonly encountered in the workplace therefore the two constructs of perceived stress and occupational burnout are interwoven.

**The Maslach Burnout Inventory** (MBI; *Hastings, Horne & Mitchell, 2004*; *Maslach & Jackson, 1981*; *Maslach, Jackson & Leiter, 1996*) is a widely accepted and utilized scale worldwide (*Coker & Omoluabi, 2009*; *Maslach & Jackson, 1981*). It has 22 items presented on a 7-point Likert scale ranging from 0 (*Never*) to 6 (*Every day*). It has three subscales: Emotional Exhaustion (EE) with items such as "I feel used up at the end of the workday," Depersonalization (DP) with items like "I treat some recipients as if they were impersonal objects," and Personal Accomplishment (PA) with items including "I have accomplished many worthwhile things in my job" (*Maslach & Jackson, 1981*). The PA subscale is positively stated thus *low* PA scores represent burned-out workers while *high* EE and DP scores represent burned-out workers.

Convergent and divergent validity has been tested against the Symptoms Checklist 90, the General Health Questionnaire and the Psychophysiological Symptoms Checklist (*Coker & Omoluabi, 2009*). The MBI is internally consistent with *Maslach & Jackson (1981)* reporting Cronbach's alphas of: EE .90, DP .79, PA .71, consistent with the Cronbach's alpha reported across several studies reviewed by *Coker & Omoluabi (2009)*. Test-retest reliability is reported at: EE .82, DP .60, and PA .80 (*Maslach & Jackson, 1981*). Cronbach's alpha in the present study was excellent: EE .94, DP .90, and PA .94.
**Demographic questions**. Participants completed demographic questions based on the Human Services Demographic Data Sheet from the MBI (e.g., sex). Participants were also asked their height and weight (from which body mass index was calculated), current employment and study commitments, education status, and income.

**Health screen**. Participants completed a pre-exercise health screen online. Participants with existing medical conditions were able to participate providing they gained an appropriate medical certificate recommending exercise participation.

**Exercise diary**. Participants reported their existing monthly exercise patterns with the following prompts: walk 5 km or more, run 3 km or more, swim laps, bicycle, attend a gym, play sport, and other. Participants responded with a frequency ranging from never to daily, and durations ranging from 0 to 30, 30 to 60, and >60 (minutes). During the two and four week questionnaires, participants reported the type, frequency and duration of exercise conducted over the study period. Participants' self-reports were monitored to ensure exercise conducted was in accordance with the allocated condition.

## Procedure

Participants were recruited through a SportUNE advertisement drive in the Tamworth and Armidale localities, through social media (Facebook), publication in the *Northern Daily Leader* newspaper, and a live interview with ABC Radio New England North West.

Participants provided electronic informed consent. Participants allocated to the exercise conditions met with the experimenter, conducted a safety brief, program overview, and blood pressure screening for hypertension. Participants in the resistance condition were taught correct exercise technique on an individual basis before being provided a training program. Cardiovascular condition training consisted of group fitness classes taken under qualified instructors, as well as individual exercise such as running, cycling, and swimming negating the requirement for individual instruction.

Participants were asked to complete a minimum of three 30 min sessions of their allocated exercise per week for four weeks based on the frequency and intensity recommendations by the American College of Sports Medicine (*Garber et al., 2011*).

The participants in the control condition were informed that they were on a waiting list for participation in exercise program, and were asked to complete the online data collection initially and at two and four weeks, before being reallocated to one of the exercise conditions.

Participants were given discounted access to SportUNE facilities in Armidale, or 360 Fitness in Tamworth both in New South Wales, Australia. Although gym membership aided in program compliance it was not a prerequisite for participation, with the training programs designed to be conducted at home with minimal equipment. Statistical comparison of participant outcomes by location did not reveal any significant differences.

## Statistical analysis and interpretation

SPSS version 21 was used for statistical analysis. For the testing of *Hypothesis 1*, participants in cardiovascular and resistance conditions were combined and recoded into a new variable *exercise* to compare exercise with control. Analysis of covariance (ANCOVA) was

**Peer**J ________________________________________________________

utilized as the test for the first and second hypotheses. Pearson correlations were calculated across all subscales to test *Hypothesis 3*. Based on previous research by *Bird (2001)* ($n = 70$) and *Gerber et al. (2013)* ($n = 12$) for a large effect size a minimum of 12 participants is needed for each condition, giving a minimum sample of 36.

# RESULTS

## Descriptive statistics

Reported monthly exercise prior to commencement of the intervention ranged from 0 to 11 hours per month ($M = 2.95$, $SD = 3.07$). An ANOVA was conducted to compare initial exercise status between conditions and was not significant ($F(2, 49) = 1.00$, $p = .38$). Participants came from a range of occupations, but most frequent was education (30.9%), government (20.0%), and medical (9.1%). All participants worked or studied for a minimum of 20 hours per week. An ANOVA was conducted to compare hours worked and/or studied between conditions and was not significant ($F(2, 21) = 1.10$, $p = .17$).

Table 1 shows baseline descriptive statistics for all subscales across treatment conditions (i.e., cardiovascular, resistance, exercise [cardiovascular and resistance combined] and control) as well as subscale change after the four weeks. Table 1 also displays *t* values that represent the comparison between the exercise and control group showing no between group differences at study enlistment but, as expected, differences on completion of the intervention.

Using *Maslach & Jackson (1981)* guidelines, participants were grouped into burnout categories low burnout (EE $\leq$ 16, DP $\leq$ 6, PA $\geq$ 39), medium (EE = 17–26, DP = 7–12, PA = 32–38), and high (EE $\geq$ 27, DP $\geq$ 13, PA $\leq$ 31). At baseline, 37.6% of the sample had high burnout, 27.9% had medium burnout, and 34.5% had low burnout, which after four weeks of exercise reduced to 14.9% high, 41.2% medium, and 43.9% low burnout. In comparison, at four weeks the control condition remained at 41.7% high, 29.2% medium, and 29.2% low burnout.

Repeated measures *t*-tests were conducted for the different conditions and were statistically significant for the exercise condition in relation to emotional exhaustion $t(31) = 3.23$ $p < .01$, large effect size (partial $\eta^2 = .59$) and personal accomplishment $t(31) = -3.29$, $p < .01$, large effect size (partial $\eta^2 = .40$), but not depersonalisation $t(31) = 1.02$ $p = .32$. Repeated measures *t*-tests for the control group did not reveal any statistically significant changes over the four weeks.

## Comparison of exercise and control

Table 2 shows the results of the analysis comparing exercise and control. The results were statistically significant for **positive well-being** and **psychological distress** with large effect sizes indicating 45% of the variance in positive well-being and 22% of the variance in psychological distress was accounted for by condition (exercise vs. control). There was greater well-being and less psychological distress in the exercise than the control condition.

A significant difference was found for **perceived stress** where exercise was found to account for 23% of the variance. Again group comparison showed that the exercise

**Table 1 Descriptive statistics and _t_ tests of the MBI, PSS, and SEES.** Initial Means (SD), mean change (SD) over the four week period, and two tailed t test (exercise, control) of the SEES, PSS, and the MBI.

| Measure | Cardiovascular ($n = 20$) | Resistance ($n = 9$) | Exercise ($n = 29$) | Control ($n = 20$) | $t$ (df) |
|---|---|---|---|---|---|
| **Positive well-being** | | | | | |
| Baseline | 15.90 (4.06) | 15.00 (3.87) | 15.62 (3.96) | 15.50 (4.39) | 0.10 (47) |
| Change over 4 weeks | 5.95 (4.18) | 6.38 (3.46) | 6.08 (3.91) | 0.10 (2.88) | 4.38 (34)[***] |
| **Psychological distress** | | | | | |
| Baseline | 9.95 (4.83) | 11.67 (6.16) | 10.48 (5.23) | 10.15 (5.52) | 0.21 (47) |
| Change over 4 weeks | −4.28 (4.45) | −4.62 (5.24) | −4.38 (4.61) | −0.50 (2.01) | −2.56 (34)[*] |
| **Fatigue** | | | | | |
| Baseline | 14.50 (6.23) | 17.00 (6.44) | 15.28 (6.29) | 17.30 (6.22) | −1.12 (47) |
| Change over 4 weeks | −2.89 (5.92) | −4.75 (4.77) | −3.46 (5.56) | −1.50 (4.90) | −0.98 (34) |
| **Perceived stress** | | | | | |
| Baseline | 18.45 (6.72) | 20.11 (7.04) | 18.96 (6.74) | 18.25 (6.20) | 0.38 (47) |
| Change over 4 weeks | −4.83 (7.25) | −6.88 (6.49) | −5.46 (6.96) | 0.50 (4.09) | −2.53 (34)[*] |
| **Emotional exhaustion** | | | | | |
| Baseline | 26.84 (11.98) | 22.50 (14.41) | 25.56 (12.62) | 22.75 (12.81) | 0.75 (45) |
| Change over 4 weeks | −6.47 (9.11) | −5.00 (9.91) | −6.04 (9.16) | 1.11 (2.26) | −2.30 (31)[*] |
| **Depersonalization** | | | | | |
| Baseline | 9.32 (6.06) | 7.88 (8.06) | 8.89 (6.58) | 6.85 (5.36) | 1.13 (45) |
| Change over 4 weeks | −0.65 (3.46) | −1.43 (5.91) | −0.86 (4.19) | −1.22 (1.92) | 0.24 (31) |
| **Personal accomplishment** | | | | | |
| Baseline | 31.84 (7.55) | 34.25 (5.47) | 32.56 (6.99) | 32.15 (7.98) | 0.26 (45) |
| Change over 4 weeks | 2.06 (4.74) | 5.86 (3.67) | 3.61 (4.72) | −1.44 (4.09) | 2.56 (31)[*] |

**Notes.**

The exercise condition is based on the combination of the cardiovascular and resistance conditions and was used as the dependent variable for the _t_ test.

[*] $p < 0.05$.

[**] $p < 0.01$.

[***] $p < 0.001$.

**Table 2 ANCOVA for MBI, PSS, and SEES, intervention vs. control.** ANCOVA results for comparison of exercise and control for key outcome measures at four weeks.

| Measure | $F$ | $df$ | Partial $\eta^2$ | Direction of differences |
|---|---|---|---|---|
| Positive well-being | 27.07[***] | 1, 33 | 0.45 | Exercise > Control |
| Psychological distress | 9.45[**] | 1, 33 | 0.22 | Exercise < Control |
| Fatigue | 2.49 | 1, 33 | 0.07 | Exercise < Control |
| Perceived stress | 9.90[**] | 1, 33 | 0.23 | Exercise < Control |
| Emotional exhaustion | 6.89[*] | 1, 30 | 0.19 | Exercise < Control |
| Depersonalization | 0.19 | 1, 30 | 0.01 | Exercise > Control |
| Personal accomplishment | 6.49[*] | 1, 30 | 0.18 | Exercise > Control |

**Notes.**

[*] $p < 0.05$.

[**] $p < 0.01$.

[***] $p < 0.001$.

condition had less perceived stress than the control. Furthermore, there were significant condition effects for the **emotional exhaustion** and **personal accomplishment**. The exercise condition had less emotional exhaustion and greater levels of personal accomplishment than the control condition.

### Comparison of cardiovascular, resistance, and control

Table 3 shows the results for the cardiovascular vs. resistance vs. control comparisons. The cardiovascular and resistance conditions both showed significantly greater positive **well-being** than the control condition, but were comparable to each other. The cardiovascular condition reduced **psychological distress** compared with the control condition. The participants in the cardiovascular and resistance conditions reported significantly less **perceived stress** than the participants in the control condition. The cardiovascular condition tended to reduce **emotional exhaustion** as compared to the control condition. The comparison for **personal accomplishment** found that the resistance condition reported significantly greater personal accomplishment than the control condition.

### Effects of exercise frequency and duration

Baseline measurements were subtracted from the four week measurements and recoded into new variables to capture change. As hypothesised, strong associations ($r > .30$) were observed between exercise conducted and change in positive well-being, and reduction in perceived stress and emotional exhaustion, see Table 4.

## DISCUSSION

Overall, the results were consistent with the assumption that exercise increases well-being whilst reducing stress and burnout. The positive health states of positive well-being and personal accomplishment were found to be strongly related to exercise. Likewise, the stress and burnout health state measures of psychological distress, perceived stress, and emotional exhaustion were found to be strongly inversely related to exercise. Also of interest was the observed absence of an association between exercise and fatigue. Cardiovascular exercise tended to be better than resistance exercise when it came to reducing psychological distress. On the other hand, resistance training was more effective in increasing personal accomplishment. Cardiovascular and resistance conditions were comparable in measured decrease in perceived stress, and emotional exhaustion. Additional exercise in excess of the minimum recommended standards may lead to further gains in positive well-being, perceived stress, and emotional exhaustion, based on correlational analysis of the change over four weeks and hours of exercise conducted.

The results largely supported *Hypothesis 1* with a large effect found for both positive well-being and personal accomplishment. Furthermore, there were large effect size reductions shown against stress and burnout including psychological distress, perceived stress, and emotional exhaustion. Contrary to predictions no effect was found for the depersonalization subscale.

The findings did not categorically support *Hypothesis 2* as the cardiovascular and resistance conditions were not directly different from one another. However, *Hypothesis 2*

**Table 3 ANCOVA for MBI, PSS, and SEES, three group comparison.** ANCOVA and pairwise comparison results for comparison of cardiovascular, resistance, and control for key outcome measures at four weeks.

| Measure | $F$ | $df$ | Partial $\eta^2$ | Direction of differences (absolute estimated marginal mean difference) |
|---|---|---|---|---|
| Positive well-being | 13.13[***] | 2, 32 | 0.45 | Cardiovascular > Control (6.27)[***] <br> Resistance > Control (6.14)[**] <br> Cardiovascular > Resistance (0.13) |
| Psychological distress | 5.30[*] | 2, 32 | 0.25 | Cardiovascular < Control (3.32)[**] <br> Resistance < Control (2.14) <br> Cardiovascular < Resistance (1.18) |
| Fatigue | 1.22 | 2, 32 | 0.07 | Cardiovascular < Control (2.66) <br> Resistance < Control (2.43) <br> Cardiovascular < Resistance (0.23) |
| Perceived stress | 4.93[*] | 2, 32 | 0.24 | Cardiovascular < Control (5.05)[*] <br> Resistance < Control (5.91)[*] <br> Cardiovascular >Resistance (0.86) |
| Emotional exhaustion | 3.35[*] | 2, 29 | 0.19 | Cardiovascular < Control (7.61)[a] <br> Resistance < Control (7.03) <br> Cardiovascular > Resistance (0.58) |
| Depersonalization | 0.31 | 2, 29 | 0.02 | Cardiovascular > Control (0.95) <br> Resistance < Control (0.15) <br> Cardiovascular > Resistance (1.09) |
| Personal accomplishment | 5.35[*] | 2, 29 | 0.27 | Cardiovascular > Control (3.47) <br> Resistance > Control (7.34)[**] <br> Cardiovascular < Resistance (3.88) |

**Notes.**

[a] $p = 0.053$.
[*] $p < 0.05$.
[**] $p < 0.01$.
[***] $p < 0.001$.

was indirectly supported as the cardiovascular vs. control and resistance vs. control outcomes showed some differences. Cardiovascular demonstrated a greater effect on psychological distress than resistance and resistance displayed a greater effect on personal accomplishment than cardiovascular when compared to the control condition. Similar effects of cardiovascular and resistance exercise were found for perceived stress and emotional exhaustion.

The correlational comparison tended to support *Hypothesis 3*, particularly with positive well-being, perceived stress, and emotional exhaustion, suggesting that there is further benefit from additional exercise in excess.

## Comparison to previous research

Results of the current study on the SEES were consistent with the reported effects found by *Cox et al. (2006)* where high intensity exercise was superior to moderate intensity exercise in increasing positive well-being. As argued by *Cox et al. (2006)*, this result has

**Table 4 Correlation matrix of MBI, PSS, and SEES change.** Correlation matrix for key measures using subscale change (over four weeks) and exercise conducted (measured in hours) ($n = 34$).

| Measure | 2 | 3 | 4 | 5 | 6 | 7 | 8 |
|---|---|---|---|---|---|---|---|
| 1. Positive well-being | −0.67[**] | −0.37* | −0.42[*] | −0.39[*] | 0.09 | 0.30 | 0.53[**] |
| 2. Psychological distress | | −0.45[**] | 0.39[*] | 0.11 | −0.10 | −0.19 | −0.22 |
| 3. Fatigue | | | 0.35[*] | 0.29 | 0.09 | −0.22 | −0.10 |
| 4. Perceived stress | | | | 0.63[**] | 0.38[*] | −0.46[**] | −0.35[*] |
| 5. Emotional exhaustion | | | | | 0.32 | −0.25 | −0.40[*] |
| 6. Depersonalization | | | | | | −0.07 | 0.21 |
| 7. Personal accomplishment | | | | | | | 0.30 |
| 8. Exercise conducted | | | | | | | |

**Notes.**

Subscale measurements are based on change on variables (i.e., four week—baseline). Exercise conducted was the number of hours over the four week study period.

[*] $p < 0.05$.

[**] $p < 0.01$.

the important implication of dispelling common advice that moderate intensity exercise is preferable to vigorous exercise for increasing positive affect (*Berger & Motl, 2000*). Also consistent with *Cox et al. (2006)*, the current study did not find an effect of fatigue in either exercise condition. This indicates that the commonly held opinion "I'm too tired to exercise" in those lacking motivation may lack credibility. A more detailed analysis of the relationship between fatigue and cardiovascular and resistance training could be gained through repeating the study using the recently developed Hecimovich–Peiffer–Harbough Exercise Exhaustion Scale (*Hecimovich, Peiffer & Harbough, 2014*).

The stress buffer hypothesis proposed by *Gerber & Puhse (2009)* and *Gerber et al. (2010)* argues that moderate exercise is the optimal stress buffer. The current findings showed inconsistent support for *Hypothesis 2* suggesting that perhaps the stress buffer relationship is more complicated than what can be explained by exercise intensity. *Lutz, Stults-Kolehmainen & Bartholomew (2010)* may assist in understanding the current inconstancies. Consistent with current support for *Hypothesis 3*, *Lutz, Stults-Kolehmainen & Bartholomew (2010)* reported an inverse relationship between stress and exercise in previously sedentary participants; however, the authors also measured participants with an established (>6 months) exercise regime finding the converse relationship where exercise increased with increasing stress. This led to the argument that those with greater motivation or self-regulation will increase exercise under stress, whereas those with less experience or motivation will exercise less under stress (*Lutz, Stults-Kolehmainen & Bartholomew, 2010*). A high fidelity study comparing previously sedentary and currently active participants across both exercise intensity and type would likely deepen understanding of the mechanisms behind the stress buffer effect.

The current ANCOVA analysis was consistent with *Gerber et al. (2013)* on the PSS and the emotional exhaustion subscale of the MBI with comparable large effect sizes. These findings suggest that exercise is an effective intervention to reduce stress and emotional exhaustion which has important organisational implications for the development of stress

and burnout interventions accounting for approximately 20% of the variance (perceived stress 23% and emotional exhaustion 19%).

The current study did not replicate the large effect size found by *Gerber et al. (2013)* on the depersonalization subscale, but found a large effect of personal accomplishment where *Gerber et al. (2013)* did find an association. *Gerber et al. (2013)* used cardiovascular exercise as the treatment, therefore as revealed in analysis of *Hypothesis 2*, the observed difference in personal accomplishment was significant only in the resistance condition. This is a logical explanation why a study measuring only cardiovascular exercise failed to record a significant effect in personal accomplishment. It is possible that the current findings in the area of personal accomplishment may be attributed to Yoga, Pilates, and Body Balance that were included in the resistance training program. In order to establish this detail, future research should ask participants to detail which forms of resistance training they conducted to allow a group wise comparison between weight training and Yoga/Pilates and other associated classes.

## Implications

With an estimated 43% of Australians suffering unhealthy levels of stress (*Nash, 2013*), it is critical that interventions targeting burnout are given due attention. Exercise is cost effective, saving $4 for every $1 invested (*Shusterman, 2010*), and a non-invasive solution to combat burnout, and can be easily implemented on a large scale. The true benefits of exercise as an intervention extend to its predisposition to target both the individual and department/organisation concurrently. Exercise is additionally beneficial as it does not require the removal of the individual from the workplace and be achieved without the stigma often associated with such a removal (*Halbesleben, Osburn & Mumford, 2006*). The preliminary nature of the findings of the present study preclude them from definitively ascertaining the exact prescription for exercise. However, the present study suggests that exercise as a whole has a noticeable positive effect on the experience of burnout, most prominently on the experience of emotional exhaustion. What the present study also suggests is that there does appear to be differences in the manner that different exercise combinations impact employee stress and burnout which the authors feel warrant further investigation as part of a larger study.

The most important outcome and recommendation from this study is that exercise is an effective intervention for reducing burnout. The proportion of the sample experiencing high levels of burnout reduced by more than half over the course of the intervention (37.6% reduced to 14.9%). The present study also suggests that there does appear to be differences in the most suitable program dependent on the unique burnout profile specific for each organisation. It is suggested that organisations that score poorly on the personal accomplishment subscale of the MBI can assist their employees by implementing a resistance training program. Alternatively, organisations wishing to emphasize psychological health and well-being concerns can concentrate on this by commencing an appropriate cardiovascular training program.

## Limitations

Limitations of the current study include its small and uneven sample size across conditions. All participants that failed to complete the program were from the resistance condition, resulting in sample inequality. The resistance condition involved more participant independence which may have resulted in lower levels of participation in comparison to the fixed times of the cardiovascular exercise classes. Participant motivation and availability to complete the training program should be considered in future studies.

This pilot study was reliant on self-report data. Whilst financially constrained in this regard, the reliance on subjective self-report entails the risk that the results are susceptible to expectancy effects and inflated self-reporting of exercise conducted. Future studies with more time and funding might address this limitation through the use of heart rate monitors (HRM). For any large scale study the authors recommend that participants use a HRM during exercise to objectively record results. Participants could subsequently upload their recorded HRM activity anonymously by linking it to the online survey response system using an ID code. This would allow greater fidelity in the measurement of intensity, and would reduce the subjectivity of self-reported exercise. Additionally, it may be possible reduce the impact of expectancy effects on the SEES, PSS, and MBI by using HRM to distract participants away from the purpose of the study.

Effect sizes were generally moderate to large but with a small sample size some small to moderate effect sizes will not be statistically significant. As a pilot study, the results indicate that research in this area is promising; however, with the limited sample size caution should be used when interpreting the results and effect sizes reported here. Well-funded studies with larger sample sizes could address this and provide confirmation of the significance of the results found here. Where resources are available, consideration should also be given to conducting a prospective longitudinal study to determine the minimum duration of exercise interventions as well as measuring and monitoring the ongoing effects of continued exercise.

## Conclusion

Burnout prevention and reduction through tailored exercise programs is a promising, cost effective, and healthy living solution for the estimated 43% of Australians experiencing unhealthy stress levels (*Nash, 2013*). The study found that exercise reduced the proportion of the sample experiencing high levels of burnout by more than half whilst also showing variances in the different aspects of stress and burnout dependent on the type of exercise conducted. With the Australian Government already promoting exercise for healthy living in the context of reducing obesity-related health problems, this research provides a valuable supplement that attests to the significant benefit of exercise to both individuals and organisations in increasing well-being, reducing perceived stress, and reducing burnout. The positive effect of resistance training on personal accomplishment and the psychological distress reducing effects of cardiovascular exercise are exciting extensions of the current literature which, if replicated, can support health and fitness professionals in developing exercise programs for optimal physical and psychological health.

## ACKNOWLEDGEMENTS

The authors would like to thank the staff at SportUNE for dedicating their time and effort encouraging participants to become involved in the present study; especially, the Chief Executive Officer David Schmude for his generous support including providing participants with complimentary access and Operations Manager Kathie Hunt who helped to advertise the study and recruit participants. We would also like to thank the Fitness Coordinator Mally McCormack and her team, who provided support, encouragement and supervision to my participants to keep them coming back to finish the program.

At 360 Fitness Tamworth we would like to thank the Owner Dwone Jones, for his generosity and support to our research, and the General Manager Nic Bird for her flexibility and support.

### Funding

There was no formal funding of this research. SportUNE and 360 Fitness (Tamworth) provided discounted membership for participants to encourage participation.

### Competing Interests

The authors declare there are no competing interests.

### Author Contributions

- Rachel Judith Bretland conceived and designed the experiments, performed the experiments, analyzed the data, contributed reagents/materials/analysis tools, wrote the paper, prepared figures and/or tables.
- Einar Baldvin Thorsteinsson conceived and designed the experiments, performed the experiments, analyzed the data, contributed reagents/materials/analysis tools, wrote the paper, prepared figures and/or tables, reviewed drafts of the paper.

### Human Ethics

The following information was supplied relating to ethical approvals (i.e., approving body and any reference numbers):

University of New England Human Research Ethics Committee (approval number HE13-051).

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
