# Peer review of "Reducing workplace burnout: the relative benefits of cardiovascular and resistance exercise"

_PeerJ, doi:10.7717/peerj.891_

## Round 0.1 · original submission · Major Revisions

· Academic Editor

Major Revisions

Burnout is a very important issue in occupational health. However, this study has major weak points pointed out by the reviewers. We think your paper is interesting and could be reconsidered after substantial revision and hope you will be able to submit a revised manuscript addressing the criticisms that have been made.

Reviewer 1 ·

Basic reporting

The basic reporting is overall adequate.

Major revisions: In the first 1-2 pages under introduction, there should be a clear purpose statement, listing the objectives of the paper. Needs a separate heading for literature review, which reads a bit thin.

Experimental design

Very small n of 49 (with 20 in control group). Wide range of ages and mix of male female. Relied on self report of exercise, which is probably not the best method (having folks report for and complete exercise may prove to be a more accurate way of assessing that exercise was completed). Major revision: I think this should be stated with some depth, perhaps in limitations section.

Validity of the findings

Good movement toward lowering burnout rates and severity of burnout, but again the n is so small....

Major Revision: I am not sure why BMI is included in this section; it reads disjunct, I would say consider removing this section.

Additional comments

I think we already know that exercise decreases stress and anxiety.

Major Revision: I would like to see greater emphasis on the burnout issue throughout the work, particularly in the results, implications, and conclusion section. In the conclusion section: What about the private sector implications around burnout and maybe even cost of burnout?

I think you are pretty close here, these changes should be fairly easy to complete, and will really enhance the scholarly level of the work.

·

Basic reporting

No Comments

Experimental design

1.In this study, participants were randomly allocated to three groups (control, cardiovascular exercise and resistance exercise), aims to distinguish the relative effectiveness against well-being, perceived stress, and burnout among three groups. Base on the purpose, it is very important of measuring exercise patterns and intensity accurately in study design. However, the author only showed participants’ exercise status but did not compare the participants’ exercise status among three groups at the beginning of study. Another concern is if the participants in different group had been polluted by another groups, for example, some of participants increased their exercise frequency or duration because the motivation of exercise were excited; or the participants of resistance exercise might increase cardiovascular exercise besides conducted resistance exercise. Pease give detailed description on how to avoid the problems above mentioned. If the author could not exclude such problem, please add some discussion about it.
2.The author just mentioned participants’ job status was similar, however the job contents and work load could affect workers psychological and physical health conditions. Please briefly describe the participants’ job contents and work load in order to prove no difference among three groups.
3.Although the number of subject is quite low but it was discussed adequately.

Validity of the findings

According to table 1, it seems that not all of the measures of Perceived Stress Scale and MBI at baseline were equal among three study groups, using data of subscale change is more suitable than using data at four weeks when comparing the effects of intervention(table 2 and 3).

Additional comments

Burnout is a very important issue in occupational health. The supporting environments in workplace (in this case exercise) are most significant fields for improvements. This is a pilot study to clarify if the pattern and intensity of exercise could get different effects on reducing burnout and perceived stress. We can see something new about the relationship of the health effects and pattern of exercise. However, this study has major weak points such as small sample size and short intervention duration, a conservative conclusion is recommended.

---

## Round 0.2 · accepted · Accept

· Academic Editor

Accept

The additional information is valuable to the reader and addresses the reviewers concerns. We look forward to seeing this interesting report in publication shortly.